# Role of β-Caryophyllene in the Antinociceptive and Anti-Inflammatory Effects of *Tagetes lucida* Cav. Essential Oil

**DOI:** 10.3390/molecules25030675

**Published:** 2020-02-05

**Authors:** Alberto Hernandez-Leon, María Eva González-Trujano, Fernando Narváez-González, Gimena Pérez-Ortega, Fausto Rivero-Cruz, María Isabel Aguilar

**Affiliations:** 1Laboratorio de Neurofarmacología de Productos Naturales, Dirección de Investigaciones en Neurociencias, Instituto Nacional de Psiquiatría Ramón de la Fuente Muñiz, Ciudad de México 14370, Mexico; albertoh-leon@hotmail.com; 2Laboratorio de Neurociencias, Instituto Nacional de Pediatría, Ciudad de México 04530, Mexico; fernandphone_1019@hotmail.com; 3Facultad de Ciencias, Universidad Nacional Autónoma de México, Ciudad Universitaria 04510, Mexico; gimena.perorte@gmail.com; 4Departamento de Farmacia, Facultad de Química, Universidad Nacional Autónoma de México, Ciudad Universitaria 04510, Mexico; joserc@unam.mx (F.R.-C.); ilaurents@hotmail.com (M.I.A.)

**Keywords:** analgesic, Asteraceae, essential oil, *Tagetes lucida* Cav., traditional medicine, β-caryophyllene

## Abstract

*Tagetes lucida* Cav. (Asteraceae) is an ancient medicinal plant commonly used to alleviate pain. Nevertheless, scientific studies validating this property are lacking in the literature. Animal models of pain were used to evaluate the antinociceptive and anti-inflammatory activities of *T. lucida* essential oil (TLEO) and a bioactive metabolite. The chemical constitution and possible toxicity of the extract and the mechanism of action of β-caryophyllene were also explored. Temporal course curves and dose–response graphics were generated using TLEO (0.1–10 mg/kg or 3.16–31.62 mg/kg) and β-caryophyllene (3.16–10 mg/kg). Metamizole (80 mg/kg) and indomethacin (20 mg/kg) were used as reference drugs in the formalin assay and writhing test in rats and mice, respectively. The β-caryophyllene mechanism of action was explored in the presence of naloxone (1 mg/kg), flumazenil (10 mg/kg), WAY100635 (0.16 mg/kg), or nitro-l-arginine methyl ester (L-NAME) (20 mg/kg) in the formalin test in rats. GC/MS analysis demonstrated the presence of geranyl acetate (49.89%), geraniol (7.92%), and β-caryophyllene (6.27%). Significant and dose-dependent antinociceptive response was produced by TLEO and β-caryophyllene without the presence of gastric damage. In conclusion, β-caryophyllene was confirmed as a bioactive compound in the *T. lucida* analgesic properties by involving the participation of receptors like opioids, benzodiazepines, and Serotonin 1A receptor (5-HT_1A_), as well as nitric oxide.

## 1. Introduction

Essential oils are natural products of aromatic plants traditionally used all over the world for thousands of years. Their effectiveness is recommended by healers and shamans throughout history in all cultures [1], for example: for treatment of infection, as an anti-inflammatory, and for relaxing or stimulating therapy [2]. Essential oils are very complex and contain several compounds, many of them in traces. The combination of all ingredients can produce the final activity of an essential oil. Nevertheless, one extracted component may be an important element in the efficacy of the entire extract of a plant [3]. In fact, extracting one component will generally not work effectively without the other trace elements to provide balance. Thus, the quality of the essential oil depends on the constituents to produce efficacy in the treatment. The chemical composition is represented by mono and sesquiterpene hydrocarbons and their oxygenated derivatives, along with aliphatic aldehydes, alcohols, and esters.

*Tagetes lucida* Cav. is an endemic Mexican ancient plant from Nahua background that is one of the most important sacred plants. Its common name varies based on the regions in which it is used for ritual, as a condiment, and for medicinal purposes, such as in baths or infusions. They include “yauhtli”, “pericón”, Saint Miguel, “mangy”, “flor de Santa María”, and “yerba anís”, among others [4]. *T. lucida* belongs to the family Asteraceae, tribe Tageteae. Some species are traditionally used for ornamental and ceremonial use due to their showy flowers [4,5]. They show up after the rainy season, and they are, in fact, a symbol of the culmination of the agricultural cycle even today in many parts of Mexico. The day “Fiesta del Pericón” is celebrated in various parts of Mexico from 28–29 September in honor of San Miguel [6]. 

Pharmacological studies of the *Tagetes* genus regarding its essential oil are infrequent; its properties are mainly described in in vitro assays in terms of antioxidant, antimycotic, anthelmintic, and insecticidal activities [7,8]. Few in vivo assays were reported; one involved its modulatory role as an antioxidant and its macular pigment levels both in serum and in the macula of healthy Swiss albino rats after chronic oral administration at two different doses [9]. The active substances of the essential oils of the medicinal plants, which are administered externally via procedures such as baths, steaming, massage, and cleaning, enter the bloodstream transdermally or through inhalation, generating benefits at different levels including the central nervous system (CNS).

Pharmacological and phytochemical studies of *Tagetes lucida* essential oil in terms of its antinociceptive and anti-inflammatory activities are lacking in literature. However, other properties were reported, such as its antibacterial, insecticidal [10,11], cytotoxic [12,13], and antioxidant activities [10]. Regarding the central nervous system (CNS), pharmacological studies reported antidepressant [14] and anxiolytic/sedative properties for this species [6]. In this study, the acute toxicity and potential analgesic-like effects of the *Tagetes lucida* essential oil and the bioactive compound β-caryophyllene were explored in two animal models of pain. Opioids, benzodiazepines, and 5-HT_1A_ receptors, as well as nitric oxide, were also examined as possible mechanisms of action.

## 2. Results

### 2.1. Phytochemical Analysis

The analyses and identification following NMR spectroscopy of the *Tagetes lucida* essential oil revealed the presence of geranyl acetate ((*E*)-3,7-dimethyl-2,6-octadien-1-ol acetate) (**1**, Figure 1; see also Appendix A) as the major compound (Table 1), representing 50% of the total content. The NMR spectrum of this essential oil showed almost exclusively the diagnostic signals for the monoterpene (**1**): δ_H_ 5.34 [1H, t, *J* = 8 Hz, H-2; δ_C_ 118.2 (C-2)], and δ_H_ 5.07 [1H, t, *J* = 8 Hz, H-6; δ_C_ 123.7 (C-6)] signals for two vinyl protons; an allylic methylene group at δ_H_ 4.59 [2H, d, *J* = 8 Hz, H-1; δ_C_ 61.37 (C-1)]; two methylene groups at δ_H_ 2.09 and 1.72 assigned to C-5 (δ_C_ 39.5) and C-4 (δ_C_ 26.26). At 2.04 ppm, a singlet (3H) was assigned to the methyl of an acetoxy group (δ_C_ 171.0 and 21.2); three other singlets (3H each) at δ_H_ 1.69, 1.67, and 1.59 were assigned to C-10, C-8, and C-9 methyl groups (δ_C_ 16.4, 25.6 and 17.6), and two quaternary vinyl signals were recorded at δ_C_ 142.2 (C-3) and δ_C_ 131.8 (C-7). Its identity was confirmed based on spectral data (NMR and MS) and by comparison with reported values in the literature (AIST: Integrated Spectral Database System of Organic Compounds [15]).

The analyses and identification following NMR spectroscopy of the *Tagetes lucida* essential oil revealed the presence of 22 compounds, representing 100% of the total content of volatiles, with 69.18% monoterpenes, 29.10% sesquiterpenes, and 1.71% non-identified compounds. The qualitative composition of volatile compounds in aerial parts was determined by GC/MS analysis (Table 1 and Figure 1; see also Appendix A), listed in order of their elution together with their retention times. The major volatile compounds were geranyl acetate (**1**, 49.89%), geraniol (**2**, 7.92%), β-caryophyllene (**3**, 6.27%), (−)-β-cubebene (**4**, 7.72%), and *cis*-β-ocimene (**5**, 5.60%). It is worth mentioning that compound **2** could be obtained via an elimination reaction of the acetoxy group from geranyl acetate using the temperature conditions in the chromatograph. In part, this could explain its lower percentage determined by GC versus the quantity determined by NMR.

### 2.2. Antinociceptive Activity in the Writhing Test

It was considered to integrate the effects of both treatments to show that the essential oil and the bioactive metabolite produced a similar profile in the temporal course (Figure 2A) and in the dose–response graph (Figure 2B). A maximal number of writhes was observed in mice receiving the vehicle; this point was reached after the first 10 min of the nociceptive agent injection (Figure 2A). The *T. lucida* essential oil (0.1, 1.0, and 10 mg/kg, i.p.) produced significant antinociceptive activity by reducing writhing behavior in a dose-dependent manner, reducing nociception (75.39%) at dosage of 10 mg/kg, i.p., in a similar fashion to β-caryophyllene at doses of 3.16 and 10 mg/kg (Figure 2A,B) (time: F_6,222_ = 29.68, *p* < 0.0001; treatment: F_6,37_ = 49.99, *p* < 0.0001; interaction: F_36,222_ = 8.68, *p* < 0.0001). Data expressed as the AUC over a period of 30 min allowed observing the significant reduction in the nociception of the essential oil and one of its metabolites, resembling that obtained with the analgesic metamizole (80 mg/kg, i.p.) (Figure 2B) (F_7,42_ = 12.52, *p* < 0.0001). 

### 2.3. Antinociceptive Activity in the Formalin Test

In this test, the temporal course curves of the shaking behavior in rats receiving the essential oil showed a significant reduction at doses of 10 and 31.62 mg/kg in neurogenic and inflammatory phases of the formalin (Figure 3A–C) (time: F_6,174_ = 34.74, *p* < 0.0001; treatment: F_4,29_ = 11.77, *p* < 0.0001; interaction: F_24,174_ = 5.23, *p* < 0.0001). No significant diminution was observed in the dosage of 3.16 mg/kg of this treatment (Figure 3A–C), whereas a significant and dose-dependent response in the neurogenic phase between doses of 10 and 31.62 mg/kg, i.p. allowed determining a half-maximal effective dose (ED_50_) = 14.54 mg/kg, i.p. (F_4,30_ = 6.12, *p* = 0.001) (Figure 3C). On the other hand, a significant but non-dose-dependent effect was obtained in the inflammatory stage (F_4,30_ = 6.02, *p* = 0.001) (Figure 3C). 

The bioactive metabolite β-caryophyllene also produced a significant reduction in the nociceptive behavior all along the temporal course (Figure 4A–C). These results allowed observing that β-caryophyllene (10 mg/kg) produced almost a 95% analgesic-like effect in the neurogenic phase (Figure 4B). Regarding the inflammatory phase (Figure 4C), β-caryophyllene at 5.62 and 10 mg/kg produced antinociception of 34% and 73%, respectively.

As observed in the temporal course curves, administration of the antagonists per se did not modify nociceptive responses in both phases of the formalin test, since behavioral nociception was observed similar to that in the vehicle group. In contrast, a differential significant effect of β-caryophyllene alone or combined with the antagonists was obtained (Figure 5A–C) (time: F_4,192_ = 105.8, *p* < 0.0001; treatment: F_9,52_ = 24.91, *p* < 0.0001; interaction: F_54,312_ = 3.65, *p* < 0.0001). The significant antinociceptive effect produced by β-caryophyllene (10 mg/kg, i.p.) was completely inhibited in the presence of NLX and FMZ, and partially inhibited when combined with WAY and L-NAME in the neurogenic phase (Figure 5A,B) (F_9,52_ = 16.48, *p* < 0.0001). Alternatively, the effect of β-caryophyllene was partially inhibited in the presence of all the antagonists tested in the inflammatory phase (Figure 5A,C) (F_9,52_ = 18.59, *p* < 0.0001).

### 2.4. Gastric Damage

The *T. lucida* essential oil (10 and 31.62 mg/kg, i.p.) (Figure 6B,C) and β-caryophyllene (3.16 and 10 mg/kg, i.p.) (Figure 6D,E) did not produce gastric damage, as observed with the anti-inflammatory drug indomethacin alone (Figure 6A).

### 2.5. Median Lethal Dose (LD_50_)

The LD_50_ was calculated as 316 mg/kg, i.p. in mice. This dosage produced mortality in 50% of mice treated with the *T. lucida* essential oil. Animals receiving 100 or 1000 mg/kg, i.p. showed a reduced ambulatory activity that was recovered after 10 and 25 min of the treatment administration, respectively. Twenty-four hours after treatment, mice administrated with 1000 mg/kg died (100% mortality), whereas those receiving 100 mg/kg showed 0% mortality. Mice surviving gained weight and did not show pathological changes in the macroscopic examination after euthanasia subsequent to the observation for 14 days.

## 3. Discussion

The Asteraceae family is endemic to the American continent. Mexico is an important center of origin and diversity for tribes, subtribes, genera, and species of this botanical family, contributing around 30% of the diversity of Asteraceae. The *Tagetes* genus belongs to the Tageteae tribe and, until now, 56 species distributed in America were reported; about 24 to 28 of them are in Mexico, but the knowledge in such species can be qualified as scanty and scattered [16]. 

*T. lucida* belongs to the Asteraceae family. It is an ancient medicinal species that, according to the literature and recent verbal reports of merchants and healers of Mexican regions, is used to treat several gastrointestinal affections involving colic and ulcer, as well as to alleviate musculoskeletal pain [6]. Because of the ethnobotanical reports for *T. lucida*, in this study, we explored the analgesic-like response using the formalin test in rats, a nociceptive pain model that allowed exploring central (neurogenic) and peripheral (inflammatory) nociception, as well as the writhing test in mice, which mimics abdominal pain.

Aromatic properties of *Tagetes* depend on the presence of essential oils. In the case of *T. lucida* Cav., the phytochemical studies of its essential oil were already reported after gas chromatography/mass spectrometry analysis. It is important to mention that the presence of methyl chavicol (estragole) was observed as a major constituent among other volatile compounds in *T. lucida* species, such as those cultivated in Guatemala (33.9%) [17], Costa Rica (95–97% aerial parts) [18], Italia (93.8% leaf and 78.2% flowers) [19], Cuba (96.8%) [10], Colombia (92.1%) [20], and Egypt (90%) [11]. However, this compound was not found in the *T. lucida* oil extracted from a species from Mexican regions, as corroborated in this study. The low or very low presence of methyl chavicol (<11.7%) agrees with other groups exploring this species [21]. 

Estragole is a natural oil constituent in several species. Studies using oral, intraperitoneal, or subcutaneous administration of this constituent reported carcinogenic effects in mice, with controversial results reported for its mutagenicity [22]. In fact, metabolites of estragole are considered stronger hepatocarcinogens due to the formation of hepatic DNA adducts in vivo and in vitro [23]. An absence of this toxic constituent in Mexican species is important to mention because of the frequent medicinal use of the plant for several ailments [6]. 

The composition of the plant oils can vary based on the genetic constitution of the species and the agronomic conditions in which the plants grow [24,25], as well as the studied part of the plant. As an example, *T. erecta* is another *Tagetes* species commonly used in Mexican traditional medicine, in addition to *T. lucida* [6,26]. Its major constituents reported in the leaf oil were piperitone (52.4%), terpinolene (11.2%), limonene (7.6%), (*Z*)-myroxide (4.2%), and piperitenone (5.0%), in comparison to the flower oil that contained piperitone (28.5%), piperitenone (10.9%), (*Z*)-myroxide (7.9%), piperitenone oxide (7.2%), β-caryophyllene (7.0%), limonene (6.9%), and terpinolene (4.7%) as the major constituents [24]. 

Another case is *T. patula*, whose major constituents in the essential oil from leaf, capitula, and total above ground herb collected in regions of India were limonene (6.2–13.6%), (*Z*)-[3-ocimene] (0.3–8.3%), dihydrotagetone (4.5–8.1%), terpinolene (0–11.2%), *p*-cymen-8-ol (3.4–11.0%), piperitone (6.1–11.9%), piperitenone (2.7–8.1%), β-caryophyllene (2.3–8.0%), and *trans*-sesquisabinene hydrate (2.0–12.5%) [27]. Thus, geographical origin plays an important role in the chemical diversity among species and even in the same species [21].

Regarding the pharmacological evaluation of the antinociceptive effects of *T. lucida* essential oil, it is not yet reported, nor is it reported for other *Tagetes* species. Nevertheless, the anti-inflammatory properties were preliminary reported in in vitro studies using *T. minuta* and *T. lucida* essential oils from Iran [8] and Colombia [20], respectively. The essential oil composition from *T. minuta* from the Iran region demonstrated dihydrotagetone (33.86%), (*E*)-ocimene (19.92%), tagetone (16.15%), *cis*-β-ocimene (7.94%), (*Z*)-ocimene (5.27%), limonene (3.1%), and epoxyocimene (2.03%). In a similar manner, this species from foothills of northern India was represented by (*E*)-ocimenone (31.8–42.2%), (*Z*)-β-ocimene (22.9–32.7%), (*Z*)-tagetone (8.0–11.0%), (*Z*)-ocimenone (6.0–10.3%), dihydrotagetone (1.7–3.7%), (*E*)-tagetone (1.0–2.5%), and limonene (1.0–1.5%) [28]. In those pharmacological studies, mechanisms of action were determined through measuring NADH oxidase, inducible nitric oxide synthase, and TNF-α mRNA expression in lipopolysaccharide-stimulated murine macrophages using real-time PCR [8] or by determination of nitric oxide, prostaglandin (PGE2) production, or TNF-α concentration [20], where *T. minuta* and *T. lucida* oils showed significant anti-inflammatory activity.

In our study, *T. lucida* essential oil produced a significant and dose-dependent pharmacological antinociceptive response in the nociceptive experimental pain model. The most abundant constituents found in this species were geranyl acetate, followed by the presence of geraniol and β-caryophyllene; minor abundance was found for (−)-β-cubebene, (*E*)-β-ocimene, γ-elemene, linalool, and β-pinene, as well as traces of others.

Geranyl acetate (100 and 200 mg/kg, i.p.) and geraniol (12.5–50 mg/kg, i.p. and 50–200 mg/kg, p.o.) were previously reported to possess significant antinociceptive activity in the writhing and glutamate tests in mice exploring antioxidant activity [29], as well as the inhibition of Na+ voltage-sensitive sodium channels [30], suggesting involvement in glutamatergic neurotransmission. No neurological alteration was produced by geranyl acetate, since motor performance was not modified in the rotarod test after at dosage of 200 mg/kg, i.p. [29]. In addition, geraniol produced antiallodynic and anti-hyperalgesic effects in neuropathic pain in rats, reducing inducible nitric oxide synthase and *N*-methyl-d-aspartate receptor 1 expression, as well as protein levels of TNF-α in the injured region [31]. 

Another interesting and important metabolite in the *T. lucida* essential oil is β-caryophyllene, which is considered a potent selective agonist of the cannabinoid receptor subtype 2 (CB2). Its antinociceptive activity in a neuropathic pain model in mice reported not only the involvement of CB2 receptors in the central nervous system (CNS) but also inhibition of p38 MAPK/NF-κB activation, as well as reducing cytokine release [32]. No psychoactive responses are associated with CNS effects of β-caryophyllene; in contrast, it is considered a neuroprotective compound [33]. In addition, it was reported as a significant inhibitor of the inflammatory phase of formalin using oral administration [34]. In our study, it was observed that opioid and benzodiazepine receptors are also involved in the antinociceptive mechanism of action of this terpene, mainly at a central level. On the other hand, in addition to these receptors, the 5-HT_1A_ serotonin receptor and nitric oxide showed partial participation in both phases of the formalin test in rats. Traditional healers recommend *T. lucida* aerial parts, not only in hot baths but also orally consumed as an infusion or hydroalcoholic extract. In a preliminary study, we explored a polar extract and the identified flavonoids and coumarins as antinociceptive compounds of this species [35]. In addition, our present data complement the potential of this medicinal species by exploring the essential oil and a bioactive metabolite, as well as likely constituents in the hot preparations of this plants. Altogether, these constituents and others might be acting in a synergistic manner to support the potential analgesic-like effect of *T. lucida* preparations in traditional medicine for alleviating pain. 

The essential oil was tested for a possible toxicological effect obtaining an LD_50_ in mice of 316 mg/kg, i.p. The results showed that *T. lucida* produced lower toxicity in comparison to other *Tagetes* species already reported, with a toxic dose of 99.6 mg/kg for *T. erecta* and 112 mg/kg for *T. patula*, per os (p.o.) [36]. 

Gastric damage is a well-known collateral adverse effect in NSAIDs because of the pharmacological mechanism of action. Indistinct inhibition of cyclooxygenase 1 (COX-1) and 2 (COX-2) by this anti-inflammatory drug is the cause of ulcers in the muscular layer of the stomach [37]. Our data agree with a preventive gastric damage instead of ulceration preliminary reported for the *T. patula* essential oil [38]. These results reinforce the medicinal use of *Tagetes* species and bioactive constituents of their essential oils for gastrointestinal diseases.

## 4. Materials and Methods 

### 4.1. Plant Material

Aerial parts of *Tagetes lucida* Cav. (Asteraceae) were collected in San Cristóbal, Municipality of Ecatepec Mexico State, Mexico. People use this plant during the time of flowering, and it is commonly collected from the field or bought in the markets. Since flowering occurs from July to December, the vegetal material for this study was collected in September 2016. Taxonomist Ana Rosa López identified the species with voucher specimen No. 82038 deposited at the Herbarium of the Biology Department of the Universidad Autónoma Metropolitana-Iztapalapa.

### 4.2. Essential Oil Preparation

The essential oil from aerial parts of *Tagetes lucida* was prepared by steam distillation for three hours using air-dried and ground aerial parts (749 g of plant material and 9.0 L of distilled water) using a modified Clevenger-type apparatus. From this procedure, 2 mL of the pure essential oil was obtained (yield 0.26%), which was dried over anhydrous sodium sulfate and stored in an amber bottle at −4 °C until further analysis.

### 4.3. Equipment and Chromatographic Conditions

NMR spectra were acquired on a Varian Unity INOVA at 400 MHz (^1^H) and 100 MHz (^13^C). The major volatile compound was identified based on NMR spectroscopy (^1^H, ^13^C, COSY, HSQC, HMBC, Varian, Santa Clara, CA, USA).

Analyses by GC/MS were carried out in an Agilent 7890 A series gas chromatograph (Agilent Technology, Palo Alto, CA, USA) coupled with a JEOL JMS-GC Mate II mass spectrometer (JEOL USA, Peabody, MA, USA). Compounds were separated on an Agilent Technology DB-5MS (30 m × 0.25 mm, film thickness 0.25 μm) capillary column, using the following GC oven temperature program: 1 min at 40 °C, followed by 8 °C/min up to 300 °C, held for 6 min. Helium was used as the carrier gas at a flow rate of 1 mL/min. The electronic ionization energy was 70 eV, and the mass range scanned was 1–600 uma. Scan rate was 1 spec/s. Transfer line and ionization chamber temperatures were 300 °C. Data acquisition and processing were performed with TSS pro 3.0 software system (Shrader Analytical and Consulting Laboratories, Detroit, MI, USA); mass spectra were compared with the National Institute of Standards and Technology-2 (NIST-2) data base. The components were identified based on the comparison of the relative retention indices calculated using a series of *n*-alkanes (C-8–C-20) and the mass spectra from the spectrometer database (NIST library), followed by comparison with published data.

### 4.4. Pharmacological Study

#### 4.4.1. Animals

Female Swiss Webster mice (25–30 g) and female Wistar rats (180–200 g) used in the study were provided by Instituto Nacional de Psiquiatría “Ramón de la Fuente Muñiz”. Animals were kept at a controlled temperature of (22 ± 1 °C) with a light/dark cycle of 12 h, and they were fed ad libitum with standard water and food. The study was carried out according to the Official Mexican Norm for the care and handling animal (NOM-062-ZOO-1999), and the international rules of care and use for laboratory animals. Experimental activity was done in the institute following the specifications issued by the Committee of Ethics and Research of the Instituto Nacional de Psiquiatría Ramón de la Fuente Muñiz with the approval of the project numbers NC-123280.0 and NC-17073.0 (CONBIOETICA-09-CEI-010-20170316).

#### 4.4.2. Reagents and Drugs

Indomethacin, Tween-80, naloxone (NLX), flumazenil (FMZ), WAY100635 (WAY), and nitro-l-arginine methyl ester (L-NAME) were purchased at Sigma (St. Louis, MO, USA). Sodium pentobarbital (SP) was acquired from PISA Farmacéutica (Mexico City, Mexico). A 37% formaldehyde solution was purchased from J.T. Baker, Mexico City, Mexico. Metamizole and indomethacin (Sigma, St. Louis, MO, USA) were used as reference nonsteroidal anti-inflammatory drugs (NSAIDs). All treatments (except WAY100635 which was subcutaneously (s.c.) injected), essential oil (3.16, 10, and 31.62 mg/kg), and β-caryophyllene (Sigma, St. Louis, MO, USA) (3.16, 5.62, and 10 mg/kg) were used in fresh preparation and intraperitoneally (i.p.) administered at a volume of 0.1 mL/10 g (mouse) or 0.1 mL/100 g (rat) body weight. The vehicle (VEH) consisted of 0.2% Tween-80 in distilled water. 

#### 4.4.3. Experimental Design

##### Antinociceptive Activity

Groups of at least six mice or rats received the vehicle or different doses of the *T. lucida* essential oil or β-caryophyllene as the possible bioactive metabolite. After 30 min, a nociceptive agent was injected to induce writhing or shaking behavior, as described below.

*Writhing test*. Abdominal nociception was induced by an i.p. injection of 1% acetic acid; this test is characterized by abdominal contractions known as writhes, described as an exaggerated extension of the abdomen combined with the outstretching of hind limbs [39]. The essential oil (0.1, 1.0, and 10 mg/kg, i.p.), β-caryophyllene (3.16 and 10 mg/kg, i.p.), and metamizole (reference drug, 80 mg/kg, i.p.) were evaluated in this test. Immediately after acetic acid administration, the number of writhes was recorded for 1 min every 5 min during 30 min. Data expressed as temporal course curves were utilized to observe changes in the maximal number of writhes induced in mice. A dose–response graph was plotted using the area under the curve (AUC) to determine the significant antinociceptive dose of the essential oil or β-caryophyllene (see timeline in Figure 7A).

*Formalin test*. After a habituation of 20 min, rats were administered with the essential oil (3.16, 10, and 31.62 mg/kg, i.p.), β-caryophyllene (3.16, 5.62, and 10 mg/kg, i.p.), or the reference drug indomethacin (20 mg/kg, i.p.). Thirty minutes later, chemical nociception was induced as follows: immediately after 50 µL injection in the sub plantar area of the right hind paw with 1% formalin by using a 30-gauge needle, each rat was placed into a glass cylinder provided with mirrors to enable a total panorama of the nociceptive behavior [40]. The number of shakings observed in the injected paw was taken as the nociceptive response. Two periods of high shaking activity were considered. The first one was present immediately after injection and lasted 5 min; this is known as the early phase (neurogenic phase). A second period was observed 5–30 min after formalin injection, denominated the late phase (inflammatory phase). Control animals received the vehicle with the same route and time of administration (see timeline in Figure 7B). The temporal course curves and dose–response graphs were plotted. The area under the curve (AUC: shaking per min) was calculated to determine the significant antinociceptive dose of the essential oil and β-caryophyllene. 

β-caryophyllene (10 mg/kg, i.p.) was also explored in the presence of the antagonists NLX (1 mg/kg, i.p.), FMZ (10 mg/kg, i.p.), WAY (0.16 mg/kg, s.c.), or the nitric oxide synthase inhibitor NG-nitro-l-arginine methyl ester (l-NAME, 20 mg/kg, i.p.). Doses were chosen from previous studies in the laboratory (see timeline in Figure 7C).

##### Gastric Damage as an Adverse Effect

Rats receiving vehicle or treatment with the reference drug indomethacin (20 mg/kg, i.p.) or the *Tagetes lucida* essential oil (10 and 31.62 mg/kg, i.p.) and β-caryophyllene (3.16 and 10 mg/kg, i.p.) alone were explored for possible gastric damage.

For gastric lesion observation, all rats were euthanized inside of a CO_2_ chamber, and then the stomachs were dissected. Then, the stomachs were filled with 10 mL of 10% formaldehyde to fix the tissue for 10 min. Finally, the stomachs were opened by the greater curvature and rinsed with distilled water to eliminate their content. A scanning of the tissue was obtained for qualitative analysis of the possible gastric ulcer lesions [41] (see timeline in Figure 7C).

##### Acute Toxicity

The method estimated the dose of the essential oil that would kill 50% of the animal population via an intraperitoneal route of administration (i.p.). Mice were treated with a maximal dosage of 2000 mg/kg, i.p. according to the indication of the Organisation for Economic Co-operation and Development (OECD) (2001). If 100% of the treated mice died, doses of 1000 mg/kg and 100 mg/kg were then tested. Mice surviving were kept under observation for the following 14 days, and their weights were registered; at the end of the study, a macroscopic evaluation was done. The median lethal dose was calculated according to the report of Lorke [42] (see timeline in Figure 7D).

##### Statistical Data Analysis

To describe nociceptive behavior (dose–response) in the writhing and formalin tests, the temporal course curves of nociception were calculated as the area under the curve (AUC) using the trapezoidal rule. Data are expressed as the mean ± standard error of the mean (SEM) of at least six repetitions. Temporal course curves were analyzed by two-way repeated-measures analysis of variance (ANOVA) followed by Tukey’s post hoc test. Dose–response data were examined by one-way ANOVA followed by Dunnett’s test or Tukey´s test using GraphPad Prism software (GraphPad Software INC, La Jolla, CA, USA), version 8.0. A *p*-value <0.05 was considered statistically significant.

## 5. Conclusions

In conclusion, the NMR and GC/MS methods allowed the analysis of *T. lucida* essential oil to demonstrate the presence of geranyl acetate as the major compound, followed by geraniol and β-caryophyllene, where the latter was confirmed as a participant bioactive metabolite in the antinociceptive activity of this *Tagetes* species. Pharmacological in vivo data gave evidence not only of the antinociceptive effects of this plant but also its safety as analgesic, since it did not produce gastric damage, unlike clinic drugs, suggesting its potential for pain therapy.

## Figures and Tables

**Figure 1 molecules-25-00675-f001:**
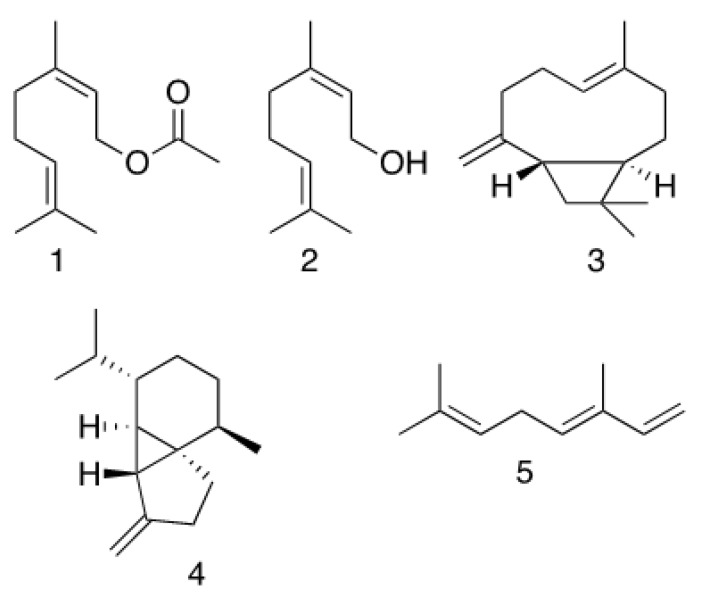
Chemical structures of geranyl acetate (**1**), geraniol (**2**), β-caryophyllene (**3**), β-cubebene (**4**), and *E*-β-ocimene (**5**) as main constituents of the *T. lucida* Cav. essential oil.

**Figure 2 molecules-25-00675-f002:**
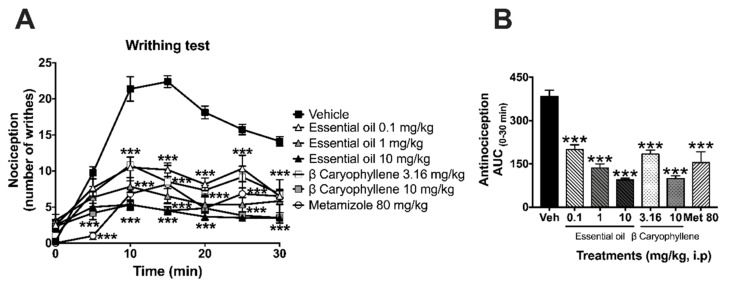
Pharmacological evaluation in the writhing test. Antinociceptive effects are expressed as temporal course curves (**A**) and dose–response (**B**) of the *Tagetes lucida* essential oil (0.1, 1.0, and 10 mg/kg) and β-caryophyllene (3.16 and 10 mg/kg) in comparison to the anti-inflammatory drug metamizole (Met 80 mg/kg) or vehicle (Veh) in mice. Two-way ANOVA followed by Tukey’s post hoc test. One-way ANOVA followed by Dunnett’s test. *** *p* < 0.001.

**Figure 3 molecules-25-00675-f003:**
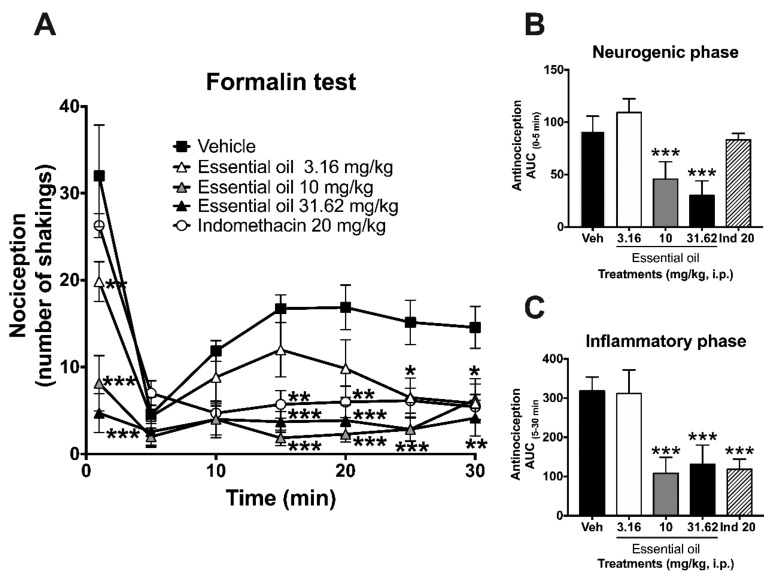
Pharmacological evaluation of *Tagetes lucida* in the formalin test. Antinociceptive effects are expressed as temporal course curves (**A**) and neurogenic (**B**) or inflammatory (**C**) dose–response of the essential oil (3.16, 10 and 31.62 mg/kg) in comparison to the anti-inflammatory drug indomethacin (Ind: 20 mg/kg) and vehicle (Veh) in rats. Two-way ANOVA followed by Tukey’s post hoc test. One-way ANOVA followed by Dunnett’s test. * *p* < 0.05, ** *p* < 0.01, *** *p* < 0.001.

**Figure 4 molecules-25-00675-f004:**
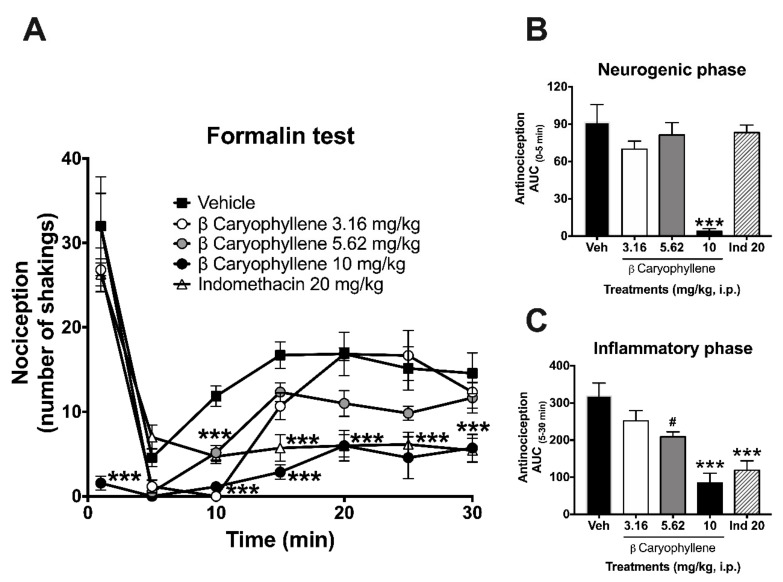
Pharmacological evaluation of β-caryophyllene in the formalin test. Antinociceptive effects expressed as temporal course curves (**A**) and neurogenic (**B**) or inflammatory (**C**) dose–response of β-caryophyllene (3.16, 5.62, and 10 mg/kg) in comparison to the anti-inflammatory drug indomethacin (Ind: 20 mg/kg) and vehicle (Veh) in the formalin test in rats. Two-way ANOVA followed by Tukey’s post-hoc test. One-way ANOVA followed by Dunnett’s test. # *p* < 0.05, *** *p* < 0.001.

**Figure 5 molecules-25-00675-f005:**
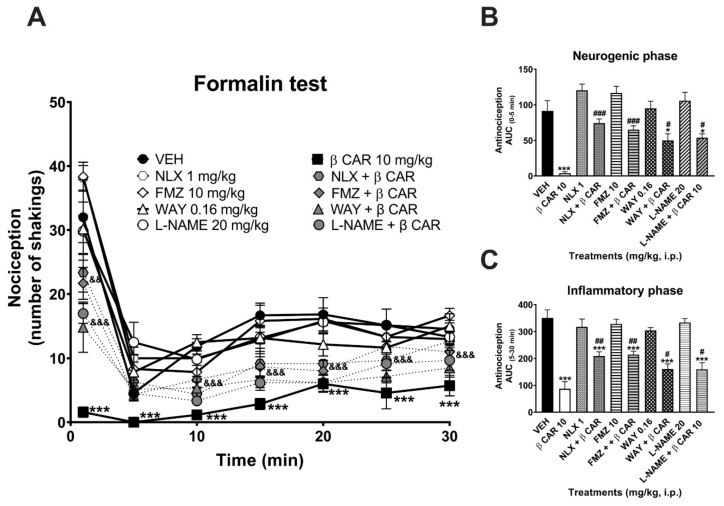
β-Caryophyllene mechanism of action. Time course curves (**A**) and dose–response in the neurogenic (**B**) and inflammatory (**C**) effects of β-caryophyllene (10 mg/kg, intraperitoneally (i.p.)) alone and in the presence of one antagonist: naloxone (NLX, 1 mg/kg, i.p.), flumazenil (FMZ, 10 mg/kg, i.p.), WAY100635 (0.16 mg/kg, subcutaneously (s.c.)), or the nitric oxide synthase inhibitor L-NAME (20 mg/kg i.p.). Each point or column is the average ± standard error of at least six repetitions. Two-way ANOVA followed by Tukey´s post hoc test. *** *p* < 0.001 vs. Veh; &&& *p* < 0.001 vs. β-caryophyllene. One-way ANOVA followed by Dunnett´s test. * *p* < 0.05, *** *p* < 0.001 vs. Veh; # *p* < 0.05, ## *p* < 0.01, ### *p* < 0.001 vs. β-caryophyllene.

**Figure 6 molecules-25-00675-f006:**
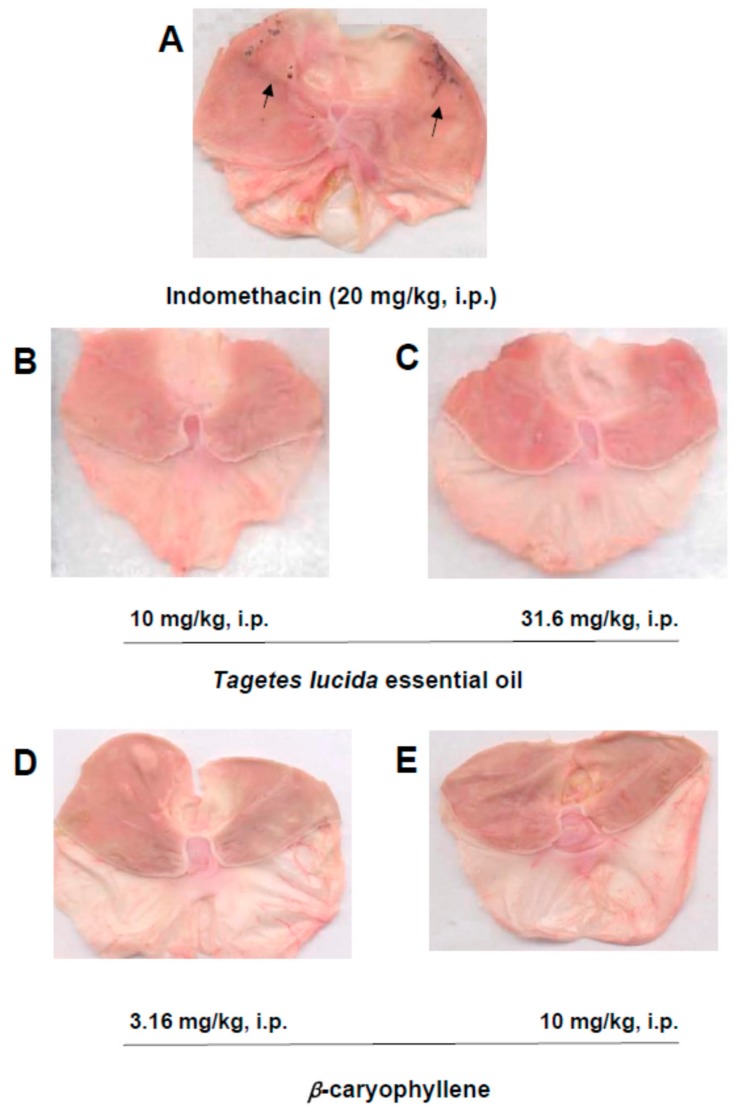
Evaluation of the gastric damage. Representative stomachs of the gastric damage explored in the presence of *Tagetes lucida* essential oil (10 and 31.6 mg/kg, i.p., (**B**,**C**), respectively) or β-caryophyllene (3.16 and 10 mg/kg, i.p., (**D**,**E**), respectively) and compared to indomethacin (20 mg/kg, i.p., (**A**)) in rats.

**Figure 7 molecules-25-00675-f007:**
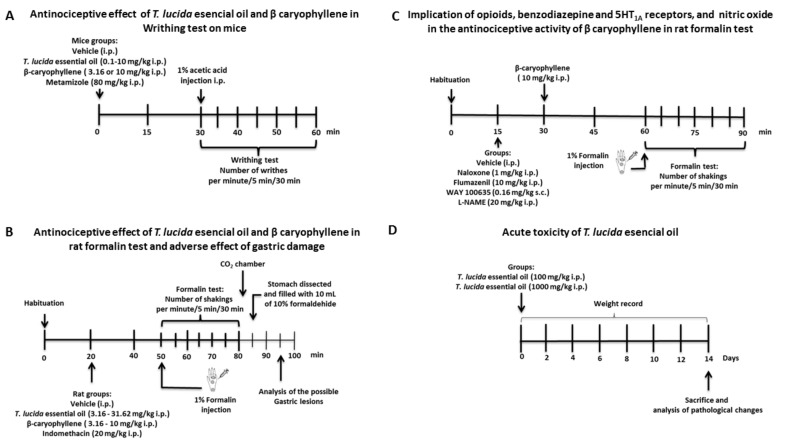
Experimental design timeline. Pharmacological evaluation of the *Tagetes lucida* essential oil and β-caryophyllene in the nociception of the writhing test in mice (**A**) and formalin test in rats (**B**). Exploration of the β-caryophyllene mechanism of action (**C**) and acute toxicity of *T. lucida* essential oil (**D**).

**Table 1 molecules-25-00675-t001:** Volatile components identified by GC/MS in aerial parts of *Tagetes lucida* Cav.

Compound ^a^	Area (%)	RI ^b^	Chemical Formula	RT (min)	Method of Identification ^c^
β-Pinene	2.16	979	C_10_H_16_	5.67	a, b
*cis*-β-Ocimene (**5**)	**5.60**	1042	C_10_H_16_	7.29	a, b
Linalool	3.61	1096	C_10_H_18_O	8.63	a, b
Geraniol (**2**)	**7.92**	1252	C_10_H_18_O	11.85	a, b
Geranyl acetate (**1**)	**49.89**	1383	C_12_H_20_O_2_	14.19	a, b, c
β-Bourbonene	0.66	1388	C_15_H_24_	14.40	a, b
β-(*E*)-Caryophyllene (**3**)	**6.27**	1455	C_15_H_24_	14.83	a, b
(*E*,*E*)-α-Farnesene	1.93	1505	C_15_H_24_	15.26	a, b
7-*epi*-α-Selinene	0.69	1522	C_15_H_24_	15.35	a, b
(−)-β-Cubebene (**4**)	**7.72**	1523	C_15_H_24_	15.80	a, b
(*E*)-iso-γ-Bisabolene	3.01	1529	C_15_H_24_	16.00	a, b
α-Cadinene	0.81	1538	C_15_H_24_	16.14	a, b
α-Muurolol	1.20	1546	C_15_H_26_O	16.31	a, b
Elemol	4.90	1549	C_15_H_26_O	17.00	a, b
Germacrene B	0.91	1561	C_15_H_24_	17.26	a, b
*E*-Farnesene epoxide	0.18	1624	C_15_H_24_O	17.66	a, b
n.i.	0.36	-	-	17.76	-
α-Cadinol	0.40	1656	C_15_H_26_O	17.94	a, b
(−)-Spathulenol	0.34	1578	C_15_H_24_O	18.05	a, b
α-Muurolol	0.73	1646	C_15_H_26_O	18.13	a, b
α-Cadinol	0.42	1655	C_15_H_26_O	18.38	a, b
14-Hydroxy-9-*epi*-(*E*)-caryophyllene	0.29	1670	C_15_H_24_O	18.59	a, b

^a^ Compounds are listed in order of elution on the DB-5 column. ^b^ Retention indices (RI) from the literature on non-polar columns reported from National Institute of Standards and Technology (NIST) library. ^c^ a: retention index; b: mass spectrum; c NMR; RT: retention time; n.i. not identified.

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
