# Peer review of "Role of β-Caryophyllene in the Antinociceptive and Anti-Inflammatory Effects of *Tagetes lucida* Cav. Essential Oil"

_molecules, 2020, doi:10.3390/molecules25030675_

Round 1
Reviewer 1 Report
In this work, González-Trujano et al. described that T. lucida essential oil from medical plant is available to alleviate pain with no gastric damage, and the authors successfully demonstrated it by in vivo assay. The results are interesting and publishable, and a minor revision is necessary.
The authors should polish the article and carefully check errors through the whole text. For example, the sentence in line 75-78 is confusing, please explain it concisely and clearly, and ‘5-HT1A’ should be ‘5-HT1A’; The names of chemical structures in Fig.2 are unnecessarily relocated in the figure. There are many abbreviations are not explained. it’s better to write a list of all of the abbreviations.Author Response
Response to Reviewer 1 Comments
Comments and Suggestions for Authors
In this work, González-Trujano et al. described that T. lucida essential oil from medical plant is available to alleviate pain with no gastric damage, and the authors successfully demonstrated it by in vivo assay. The results are interesting and publishable, and a minor revision is necessary.
Point 1. The authors should polish the article and carefully check errors through the whole text. For example, the sentence in line 75-78 is confusing, please explain it concisely and clearly, and ‘5-HT1A’ should be ‘5-HT1A’;
Point 2. The names of chemical structures in Fig.2 are unnecessarily relocated in the figure.
Point 3. There are many abbreviations are not explained. it’s better to write a list of all of the abbreviations.
ANSWERS
Thank you for your time and observations. As recommended by the reviewer:
Point 1. Misspelling and English redaction were checked again in the entire document. The sentence in line 75-78 (now 87-90) was rewritten to be clearer and concise. Abbreviation of 5-HT1A receptor was homogenized in the entire document.
Point 2. Duplicated names of chemical structures in Fig. 2 were removed.
Point 3. A list of abbreviations was included.

Reviewer 2 Report
The study entitled “β-caryophyllene involved in the antinociceptive and anti-inflammatory effects of Tagetes lucida Cav. Essential oil “ submitted to Molecules by Alberto Hernandez-Leon and co-workers describes the effect of Tagetes lucida Cav. (Asteraceae), an ancient medicinal plant commonly used to alleviate pain. β-caryophyllene was a major component and confirmed as a bioactive compound in the T. lucida analgesic properties by involving participation of receptors like opioids, benzodiazepines and 5-HT1A, as well as nitric oxide.
The manuscript focuses an interesting topic that is worth to be published, but in my opinion after minor revision .
Moderate editing of English language is required to make the article more understandable.
Author Response
Response to Reviewer 2 Comments
Comments and Suggestions for Authors
The study entitled “β-caryophyllene involved in the antinociceptive and anti-inflammatory effects of Tagetes lucida Cav. Essential oil “ submitted to Molecules by Alberto Hernandez-Leon and co-workers describes the effect of Tagetes lucida Cav. (Asteraceae), an ancient medicinal plant commonly used to alleviate pain. β-caryophyllene was a major component and confirmed as a bioactive compound in the T. lucida analgesic properties by involving participation of receptors like opioids, benzodiazepines and 5-HT1A, as well as nitric oxide.
The manuscript focuses an interesting topic that is worth to be published, but in my opinion after minor revision.
Point 1. Moderate editing of English language is required to make the article more understandable.
ANSWERS
Point 1. Thank you for your time and comments. Misspelling and English redaction were checked again in the entire document.

Reviewer 3 Report
This paper refers to the analysis of essential oil and bioactive metabolites from Tagetes lucida, a medicinal plant, from the point of view of the the antinociceptive and anti-inflammatory activities. The GC/MS analysis of the essential oil was also discussed.
Results seems to be interesting for readers, taking into account the antinociceptive and anti-inflammatory activities of essential oil from medicinal plants, maybe less harmful for humans.
In general, the manuscript is well written and accurate in its development but some revisions are required in order to improve it.
The bibliography used is relevant and updated. In fact, the antinociceptive activity of T. lucida alcoholic extract was also explored by the authors ( Life Sciences 231 (2019) 116523) and I did not understand why the authors did not did not refer to this paper in the manuscript.
Concerning the GS/MS and RMN analysis, in the non published data the TIC chromatogram, provides multiple eluted analytes, the majority identified compounds being indicated in table 1. However, NMR spectra of the oil sample would be useful too, to understand how to assign NMR signals from a complex sample, such as an essential oil.
The authors should emphasize the novelty of the conducted study and indicate more clearly the results from the study in the conclusions.
Apart from that, I consider the paper suitable for publications after minor revision.
Author Response
Response to Reviewer 3 Comments
Comments and Suggestions for Authors
This paper refers to the analysis of essential oil and bioactive metabolites from Tagetes lucida, a medicinal plant, from the point of view of the the antinociceptive and anti-inflammatory activities. The GC/MS analysis of the essential oil was also discussed.
Results seems to be interesting for readers, taking into account the antinociceptive and anti-inflammatory activities of essential oil from medicinal plants, maybe less harmful for humans.
In general, the manuscript is well written and accurate in its development but some revisions are required in order to improve it.
Point 1. The bibliography used is relevant and updated. In fact, the antinociceptive activity of T. lucida alcoholic extract was also explored by the authors ( Life Sciences 231 (2019) 116523) and I did not understand why the authors did not did not refer to this paper in the manuscript.
Point 2. Concerning the GS/MS and RMN analysis, in the non published data the TIC chromatogram, provides multiple eluted analytes, the majority identified compounds being indicated in table 1. However, NMR spectra of the oil sample would be useful too, to understand how to assign NMR signals from a complex sample, such as an essential oil.
Point 3. The authors should emphasize the novelty of the conducted study and indicate more clearly the results from the study in the conclusions.
Apart from that, I consider the paper suitable for publications after minor revision.
ANSWERS
Thank you for your time and comments.
Point 1. The reference from a preliminary study of T. lucida already reported in Life Sciences (2019) was included in discussion of the manuscript, whereas results were emphasized in the conclusion of this study.
Point 2. As suggested, figures showing the 1H-NMR and 13C-NMR of the oil sample were included in the supplementary material.
Point 3. Traditional healers recommend T. lucida aerial parts not only in hot baths but also orally consumed as an infusion or hydroalcoholic extracts. In a preliminary study, we explored a polar extract and the identified flavonoids and coumarins as antinociceptive compounds of this species. In addition, our present data complements the potential of this medicinal species by exploring the essential oil and a bioactive metabolite also likely constituents in the hot preparations of this plants.
Conclusion was clarified about the results obtained in this study.

Reviewer 4 Report
In this study, the acute toxicity and potential analgesic effect of T. lucida essential oil in two animal models of pain are reported; moreover, the phytochemical profile of the essential oil, by GC/MS analysis, was determined. The involvement of β-caryophyllene, a bioactive compound contained in the essetial oil, in the antinociceptive and anti-inflammatory effects of the essential oil was described.
The study is well conducted.
Some suggestions are reported below:
The same authors published in 2019 the article entitled “Identification of some bioactive metabolites and inhibitory receptors in the antinociceptive activity of Tagetes lucida Cav.” By Gonzalez-Trujano Life Sciences 231 (2019) 116523. I suggest to insert this reference and to discuss it in the article.
The botanical name of the plant should be in italics in all the manuscript.
The fig 3A is confused; should be better divide this figure in two figures.
Author Response
Response to Reviewer 4 Comments
Comments and Suggestions for Authors
In this study, the acute toxicity and potential analgesic effect of T. lucida essential oil in two animal models of pain are reported; moreover, the phytochemical profile of the essential oil, by GC/MS analysis, was determined. The involvement of β-caryophyllene, a bioactive compound contained in the essetial oil, in the antinociceptive and anti-inflammatory effects of the essential oil was described.
The study is well conducted.
Some suggestions are reported below:
Point 1. The same authors published in 2019 the article entitled “Identification of some bioactive metabolites and inhibitory receptors in the antinociceptive activity of Tagetes lucida Cav.” By Gonzalez-Trujano Life Sciences 231 (2019) 116523. I suggest to insert this reference and to discuss it in the article.
Point 2. The botanical name of the plant should be in italics in all the manuscript.
Point 3. The fig 3A is confused; should be better divide this figure in two figures.
ANSWERS
Thank you for time and comments.
Point 1. We are including the reference and discussion of the preliminary study of a polar extract and metabolites of this species that complement our recent results with the essential oil of T. lucida.
Point 2. We check that botanical name is homogeneously written in italics in the entire document.
Point 3. It was considered integrate the effects of both treatments in figure A, to show that the essential oil and the bioactive metabolite produced a similar profile in the temporal course and in the dose-response graphic. This point has been justified in the text.
